# Thrombogenic Risk Induced by Intravascular Mesenchymal Stem Cell Therapy: Current Status and Future Perspectives

**DOI:** 10.3390/cells8101160

**Published:** 2019-09-27

**Authors:** Louise Coppin, Etienne Sokal, Xavier Stéphenne

**Affiliations:** Laboratoire d’Hépatologie Pédiatrique et Thérapie Cellulaire, Unité PEDI, Institut de Recherche Expérimentale et Clinique, Université catholique de Louvain (UCLouvain), 1200 Brussels, Belgium; etienne.sokal@uclouvain.be (E.S.); xavier.stephenne@uclouvain.be (X.S.)

**Keywords:** cell- and tissue-based therapy, mesenchymal stem cells, thrombosis, anticoagulants

## Abstract

Mesenchymal stem cells (MSCs) are currently studied and used in numerous clinical trials. Nevertheless, some concerns have been raised regarding the safety of these infusions and the thrombogenic risk they induce. MSCs express procoagulant activity (PCA) linked to the expression of tissue factor (TF) that, when in contact with blood, initiates coagulation. Some even describe a dual activation of both the coagulation and the complement pathway, called Instant Blood-Mediated Inflammatory Reaction (IBMIR), explaining the disappointing results and low engraftment rates in clinical trials. However, nowadays, different approaches to modulate the PCA of MSCs and thus control the thrombogenic risk after cell infusion are being studied. This review summarizes both in vitro and in vivo studies on the PCA of MSC of various origins. It further emphasizes the crucial role of TF linked to the PCA of MSCs. Furthermore, optimization of MSC therapy protocols using different methods to control the PCA of MSCs are described.

## 1. Introduction

Solid organ transplantation is still the treatment of choice for patients presenting end-stage organ disease. However, due to organ shortage, the growing demand for organs and the substantial risks of morbidity and mortality linked to immunosuppression and surgery, other treatments, such as cell-based therapy, are being developed. The aim is that the infused cells migrate to the injured tissue to regrow, repair, or replace the diseased cells. Although mature cells were first investigated, nowadays there is a growing interest for the use of mesenchymal stromal cells (MSCs), because they can easily be isolated and expanded in vitro, differentiate into various cell lineages and present immunomodulatory properties. However, some concerns have been raised regarding their safety and their compatibility with blood [1,2]. Activation of coagulation has indeed been reported in several transplanted patients [3,4,5,6,7,8], and could be linked to tissue factor (TF) expression. Some even describe a simultaneous activation of the complement pathway, called the Instant Blood-Mediated Inflammatory Reaction (IBMIR), which could explain the low engraftment rates of the transplanted cells.

To further optimize MSC transplantation, the interaction between the transplanted cells and the injection site, often being the blood circulation, needs to be well understood. Understanding the mechanism offers the opportunity to develop novel targets/strategies to prevent the activation of both the coagulation and complement pathways. Therefore, in this review, we first summarize briefly the concept of hemostasis, the effect of TF and the cross-talk between coagulation and inflammation. Next, we discuss current knowledge about the interaction between transplanted cells and blood, specifically focusing on the thrombogenic risk induced by MSCs. We discuss the link between coagulation and inflammation through the IBMIR. Finally, we review the different methods that groups investigated to modulate this thrombogenic risk and improve MSCs therapy.

## 2. Hemostasis

Hemostasis is the process that maintains intravascular blood in a fluid state and prevents extensive hemorrhages or thrombi from occurring after vascular damage. Hemostasis consists of three phases, including, first, the primary hemostasis, which closes the vascular breach by forming a platelet plug. Next, to consolidate the clot, fibrin strains are formed by activation of the coagulation. Finally, fibrinolysis resorbs and prevents the clot from spreading to adjacent normal areas.

### 2.1. Platelet Activation

Recent studies showed that platelets could be activated by two distinct pathways, acting in parallel or separately. Firstly, by subendothelial collagen, exposed after a vascular injury, or secondly by generated thrombin, due to activation of the coagulation by TF. The adhesion of platelets to the site of injury is due to the interactions in the collagen pathway between the platelet glycoprotein VI with the exposed collagen and platelet glycoprotein Ib-V-IX with the collagen-bound von Willebrand factor (vWF). Platelet activation in this pathway is independent of thrombin generation [9,10]. In the thrombin-initiated pathway, TF plays a key role in the activation of platelets. TF forms—with factor VIIa—an essential enzymatic complex, factor VIIa-TF, which triggers by activating factor IX, a proteolytic cascade, generating thrombin [11]. Thrombin, which is the strongest platelet agonist, then cleaves the protease-activated receptor (PAR) on the platelet surface, activating the platelets. Once activated, platelets change shape, release the contents of their α and dense granules, containing agonists such as adenosine diphosphate (ADP), thrombin and thromboxane A2, which, in turn, can activate other platelets and upregulate their integrin adhesion receptors. The most important being glycoprotein IIb/IIIa, that binds to fibrinogen or vWF, which leads to platelet aggregation [12]. This further enhances the recruitment of other platelets to the vascular breach, allowing the formation of a stable thrombus.

Platelets play an essential role in blood hemostasis, not only by forming platelet thrombi, assuring primary hemostasis, but their activation is also essential for enhancing the coagulation. Inactivated platelets express acidic phospholipids, such as phosphatidylserine [13], which express a negatively charged pole directed inwards. A specific enzyme system called flippase, prevents the spontaneous reversal to the outside of this negative pole. When platelets are activated, the increase of cytosolic calcium inhibits this flippase and the negatively charged pole of PS is flipped to the surface of the cell. This negatively charged surface plays a critical role in the amplification of prothrombin cleavage into thrombin and in the release of the content of alpha and dense granules, containing different components critical for thrombus formation [14,15].

### 2.2. The Coagulation

The main aim of coagulation is the generation of fibrin by activating inactive soluble factors through a complex proteolytic cascade. This reaction is intended to be localized and occurs only where the platelet plug was formed. A localized activation is obtained by two methods. First, the activation of these chains of reactions, converting fibrinogen to fibrin, is the most efficient and explosive when the reactions occur at the surface of activated platelets in the platelet plug. Second, several coagulation inhibitors limit the activation of coagulation to the site of injury and platelet deposition. These inhibitors include antithrombin II (ATIII), Tissue Factor Pathway Inhibitor (TFPI) and the thrombomodulin pathway by activation of protein C [16].

Previously, coagulation was described as two coagulation pathways, an extrinsic and an intrinsic pathway, converging into one common pathway. Nowadays, coagulation is considered as one big chain of reactions with multiple interactions between the coagulation factors. Nevertheless, for clinical investigations, this compartmentalization is still used, with laboratory tests studying one or the other pathway. The prothrombin time (PT) or the better-known International Normalized Ratio (INR) studies the extrinsic pathway, including factor II, V, VII, X and fibrinogen. The activated partial thromboplastin time (aPTT) or activated clotting time (ACT) studies the intrinsic pathway, including factor II, V, VIII, IX, X, XI, XII, and fibrinogen. These tests, nevertheless, will only become abnormal if the concentrations of clotting factors are <50% are even <30%–40% for some factors. In contrast, only 25%–50% of clotting factors are required for correctly functioning coagulation [17].

Activation of coagulation can be described in three phases (Figure 1):Initiation phase: Coagulation is activated by TF through the extrinsic pathway, forming, with factor VIIa, an essential enzymatic complex, factor VIIa-TF. This complex is the fuse that triggers coagulation, through activation of factor IX and X, generating only a small amount of thrombin (factor IIa). This initiation phase, nevertheless, is inefficient because pro-cofactor V and VIII are not yet in their most active form.Amplification phase: The small amount of thrombin formed during the initiation phase induces an important amplification loop via two distinct mechanisms. Thrombin increases platelet activation and adhesion through platelet agonist properties and induces the conversion of cofactor V and VIII in their active form, Va and VIIIa. Factor V is released from the alpha granules and factor VIII is released from its carrier, vWF, to be activated on the negatively charged platelet surface. Thrombin activates factor IX as well, present on the platelet surface, which forms, with cofactor VIIIa, a tenase complex (IXa-VIIIa), which converts inactivated factor X into its active form. In turn, activated factor Xa forms, with cofactor Va, the prothrombinase complex (Xa-Va), which converts prothrombin [18] into a large amount of thrombin (IIa), leading to an efficient burst of all enzymatic reactions of the whole cascade.Propagation phase: Activation of platelets and the intrinsic pathway leads to the formation of massive amounts of thrombin, which is the principal driving force of coagulation. In turn, these vast amounts of thrombin lead to the production of large quantities of fibrin strains—the final product of coagulation—which consolidates the blood clot [11,19,20,21].

## 3. Tissue Factor

Tissue factor is a transmembrane protein, which exists in two forms, membrane-bound and free TF. Membrane-bound TF is present on different cell types, fibroblasts, pericytes of the adventitia and medial smooth muscle cells in the vessel wall. After chemical stimuli, TF is also expressed on endothelial cells and monocytes. First, endothelial cells were thought to act as a barrier between factor VIIa in flowing blood and cellular sources of TF in the vessel wall, preventing, in this way, the initiation of coagulation in the absence of injury. Yet free TF can be found in circulating blood. Free TF is present as soluble TF, resulting from a proteolytic cleavage at or near the transmembrane domain of the membrane-bound TF, and in microparticles, arising from monocytes and macrophages membrane lipid rafts [22,23]. The circulating TF bearing microparticles alone do not seem to initiate coagulation effectively. However, when they bind and fuse with activated platelets via a mechanism involving P-selectin glycoprotein ligand-1 on the microparticles and P-selectin on the platelets, they initiate coagulation [11]. TF exists in an inactive encrypted form and an active form that can initiate blood coagulation. The exact underlying mechanism of decryption is uncertain, and different models have been proposed, such as dimer formation, lipid reorganization, phosphatidylserine [13] exposure and formation of disulphide bonds [24]. Besides the initiation of blood coagulation, TF also has other functions in adhesion, migration, inflammation, and cell signaling, which are important for angiogenesis and tumor progression [25]. TF expression in cancer cell lines promotes hematogenous tumor dissemination and is essential for tumor angiogenesis and growth. By generating thrombin and thus fibrin, TF facilitates tumor–cell interactions with host vascular cells, platelets, and endothelial cells, promoting tumor metastasis [23].

## 4. Cross-Talk Pathways between Coagulation and Inflammation

There is, in fact, a major cross-talk between coagulation and inflammation, which optimizes the organism’s response to injury and invasion by pathogens. Whereby the activation of one system may increase the other, resulting in a wide range of diseases that may cause varying degrees of excessive inflammation and/or thrombosis. First, there is the central player and common trigger, TF. In response to various injurious stimuli, the expression of TF is rapidly upregulated and de-encrypted by circulating monocytes and perivascular cells, exposing TF to blood. This allows TF to complex with circulating factor VIIa, generating thrombin, as described in Section 2.2., which, in turn, induces platelet activation and PAR-mediated cell signaling, promoting a myriad of pro-inflammatory events. These include, recruitment and activation of neutrophils, platelets and monocytes, release of proinflammatory cytokines (TNFα, interleukin 1β, 6 and 8, …) and activation of the complement system.

The complement system is an essential component of the innate immune response and, similarly to the coagulation system, it is composed of numerous proteins working as a proteolytic cascade. This system can be activated through three pathways (classical, lectin and alternative pathway), that converge in the formation of two C3 convertases, which catalyzes cleavage of C3 into C3a and C3b, and subsequently, C5 into C5a and C5b when the level of C3b reaches a critical threshold. The anaphylatoxins C3a and C5a, mediate many inflammatory and prothrombotic processes by binding to their protein-coupled receptors. Both C3a and C5a are crucial to the recruitment and activation of innate immune cells such as monocytes, neutrophils, and macrophages and to inducing changes in endothelial permeability. C5a enhances coagulation by inducing TF expression on neutrophils and endothelial cells and vWF secretion by endothelial cells.

Other key molecular interactions have been described as well, involving protein C, microparticles, neutrophil extracellular traps, plasmin(ogen)/fibrinolysis and the contact activation pathway. These mechanisms highlight the complex relationship between coagulation and inflammation, and although in the last few years great advances have been made to understand this relationship, there is still much to learn and discover. Novel studies should provide innovative strategies and targets to simultaneously block prothrombotic events and inflammation, without harming hemostasis [26].

## 5. Cell-Based Therapy

Cell transplantation is a fast-developing therapy in regenerative medicine. The aim is to regrow, repair or replace damaged or diseased cells, organs or tissues with the infused cells. Cell-based therapy, compared to whole-organ transplantation, is a less invasive procedure with lower morbidity/mortality rates. The risk related to the surgical procedure is significantly reduced and when the function of the graft is lacking, or when long-term graft loss occurs, the native organ is still present. In addition, cell-based therapy does not prevent the possibility of whole-organ transplantation later. However, using mature cells for this therapy is limited due to the inability of these cells to proliferate in vitro, due to the metabolic damages induced by cryopreservation and to organ shortage as well [27,28,29].

Other cell sources, such as mesenchymal stem cells (MSC), from various tissues/organs are currently under evaluation in numerous clinical trials. The most widely studied sources of MSCs include bone marrow, adipose tissue, umbilical cord, fetal tissue, placenta and amniotic fluid. In 2006, the International Society for Cellular Therapy proposed minimal criteria to define human MSCs [30]. MSCs are multipotent cells that must be plastic-adherent in standard culture conditions, must express differentiation markers such as CD105, CD73, and CD90 and lack expression of CD45, CD34, CD14, or CD11b, CD79alpha, or CD19 and HLA-DR surface molecules and must differentiate into adipocytes, osteoblasts and chondroblasts in vitro.

MSCs are fibroblast-like cells that can be isolated and expanded efficiently in vitro. They do not express tissue-specific characteristics, but they can differentiate into specialized cells, expressing a different phenotype than their precursor, under the influence of specific signals. Some suggest that MSCs in adult tissues are niches of reparative cells, that can be mobilized and differentiated in response to wound signals from the injured tissue [31].

Their differentiation potential has been widely explored in preclinical and clinical studies. The fundamental principle is that undifferentiated MSCs, after transplantation, could migrate to the site of injury and differentiate under the influence of local environmental signals, into cells expressing the appropriate phenotype. Once differentiated, these cells could then contribute to the repair of the injured tissue. Encouraging results have been obtained in a broad spectrum of disorders, including cardiovascular repair, treatment of lung fibrosis, spinal cord injury, cartilage repair [31], and acute liver failure [7]. In different studies, it was discovered that MSCs do not only have a differentiation potential but can also have a regulatory effect on the immune microenvironment, providing optimal conditions for tissue regeneration. It is clear, nowadays, that the immune system plays a key role in MSCs-mediated tissue regeneration. Through regulation of the immune cells, inflammation can be reduced in areas of tissue damage, and disease progression in chronic tissue injury can be hampered. MSCs can regulate or inhibit immune cell activity, either through secretions (paracrine effect) or direct cell contact. The interaction between MSCs and immune cells, however, is complex and still requires further experimental investigations [32,33].

Due to their immunomodulatory and regenerative properties, MSCs can potentially be used as a novel therapy for many diseases. MSCs can, for example, be used for the treatment of liver diseases, which is a major health concern with important morbidity and mortality. In general, all types of chronic hepatitis will—without an efficient treatment—progress to end-stage liver disease, for which the only curative treatment is liver transplantation [34]. However, due to organ shortage, other therapies, such as liver MSCs transplantations, are currently being investigated. The immunomodulatory and antifibrotic properties of different types of MSCs were then investigated to treat liver cirrhosis [34] and acute liver failure [35]. Recently, encouraging results were obtained using liver-derived MSCs in cirrhotic patients with acute-on-chronic liver failure or acute decompensation, inducing a significant MELD and bilirubin improvement [36].

MSC therapy involves the transplantation of either autologous or allogeneic MSCs into patients, through local (direct injection in the tissue of interest) or diffuse (intravenous (IV) or intra-arterial injection) administration. The injection site depends on the desired therapy of the cell-based transplantation. IV injection is the most widely used because it is less invasive, can be repeated easily and cells remain close to the oxygen- and nutrient-rich vasculature after extravasation into the tissue of interest. Even if there is ample evidence of MSC homing after systemic infusion, this process still needs to be optimized because only a small fraction of transplanted MSCs reach the targeted tissue. This can partially be explained because infused MSCs are trapped in the lungs immediately after systemic administration and subsequently cleared and distributed to other tissues. This can be avoided when cells are infused directly near to the tissue of interest. For example, to treat liver diseases, MSCs can be infused in the portal vein with higher engraftment rates [37]. Encouraging results were obtained in patients presenting inborn metabolic liver diseases [5,38]. Additional studies are ongoing to further elucidate the homing capacities of MSCs and the factors influencing this process, allowing scientists to optimize their migration capacities and their therapeutic effects in the targeted tissue in the future [39].

Moreover, in recent years, the therapeutic use of MSCs has been hampered with variable or even conflicting results due to different reasons. First, their self-renewal and differentiation potential is affected by the age of the donor, genetics, and exposure to environmental stress. Their proliferation and differentiation potential are variable and decrease further upon passaging in vitro. Second, obtaining the donor’s tissue/organ can sometimes involve invasive or painful procedures with possible donor site morbidity. Other alternative sources such as perinatal (umbilical cord, umbilical cord blood, and placenta) or fetal sources (amniotic fluid or fetal tissue) are considered to be a more desirable tissue source [40]. Finally, many problems still have to be solved to standardize the process of MSCs transplantation, including the optimal timing, dose and route for transplantation, improvement of the engraftment and survival rate of the transplanted cells and the safety during MSCs transplantation.

## 6. Procoagulant Activity

### 6.1. Islet and Hepatocyte Transplantation

Thrombogenesis induced by cell transplantation was described for the first time during clinical trials for type 1 diabetes. Portal vein thrombosis induced by pancreatic islets transplantations became a well-known and feared complication [41,42]. However, nowadays, by using anticoagulants such as heparin, the safety of these infusions has massively improved [43,44]. Bennet et al. described first in vitro, and later in vivo, the thrombotic reaction when pancreatic islets were in contact with blood, which they called Instant Blood-Mediated Inflammatory Reaction (IBMIR) [45]. IBMIR has been described as a dual activation of the coagulation and the complement pathway by islets or cells bearing TF. In vitro, in a whole-blood Chandler tubing loop model, islets induced a significant decrease in both platelets and polymorphonuclear neutrophils (PMNs), and an increase in C3a levels. To further understand the underlying mechanism of IBMIR, Liuwantare et al. [46] studied the coagulation and complement activation induced by porcine neonatal islets cell clusters (NICC) in platelet-rich (PRP) and platelet-poor plasma (PPP). They showed that xenogeneic IBMIR was characterized by rapid, platelet-independent thrombin generation. Platelets, however, both accelerated and exacerbated this reaction. Confirming that both pathways were activated during IBMIR. They were the first to show that PMNs activation during IBMIR is a complement-dependent mechanism. They highlighted that the activation and the recruitment of PMNs to transplanted islets represent an important therapeutic target to prevent IBMIR. Because IBMIR induced an estimated 40% loss of cell function in NICC, with major mitochondrial damages in the surviving cells. In vivo, intraportal infusion of islets in a porcine model induced numerous blood clots in portal veins, containing, occasionally, entrapped islets [45,47]. In a clinical trial, 24 patients with chronic pancreatitis underwent autologous islet transplantation after total pancreatectomy. Patients receiving the highest islets dosage showed a significant decrease in platelets; however, no difference has been observed in C3a levels [48]. Cabric et al. [49] attempted to prevent IBMIR by applying surface-immobilized functional heparin to islets, trying to mimic the protective anticoagulant activity normally present at the endothelial lining of the vascular wall. They showed that the entire cell surface of the cells could be covered with heparin and did not interfere with the function of the islets. Identical insulin secretion and engraftment rates were observed in a syngeneic islet transplantation model in diabetic mice. Others successfully reduced IBMIR by adding a thrombin inhibitor, Melagatran [50], which activated protein C synergistically with a platelet glycoprotein IIb-IIIa inhibitor, Tirofiban [51], N-acetyl-L-cysteine [52], or even α-1 antitrypsin [53].

IBMIR could be an explanation for the remarkable difference in success rate between pancreas transplantation and islets transplantation, despite the same histocompatibility barrier and immunosuppressive treatment. Dual activation of the coagulation and complement pathway causes rapid encapsulation of the transplanted cells in a blood clot, with infiltration by activated PMNs, leading to early cell destruction [13,45,48,54,55,56]. TF seems to play a key role in triggering IBMIR, because this reaction could be antagonized in a Chandler tubing loop model by blocking the TF pathway using monoclonal antibodies or site-inactivated factor VIIa [47]. IBMIR has also been described in hepatocytes transplantation. Adding hepatocytes in a whole blood loop model induced a significant decrease of platelets, PMNS and an increase in C3a levels after 1 hour [57]. Recently, the same group showed that the expression of vWF on the surface of isolated hepatocytes could induce, in contact with blood, adhesion of platelets, forming an ideal surface for coagulation. Once thrombin is generated, the thromboinflammatory reactions will then be strongly amplified and cause rapid destruction of the exposed cells [58].

Different strategies have been investigated to prevent IBMIR after hepatocytes transplantation. Gustafson et al. [59], compared the capacity of low-molecular-weight dextran sulfate (LMW-DS) to conventional anticoagulant therapy, unfractionated heparin, to control IBMIR in vitro in a whole-blood Chandler loop model. They showed that LMW-DS controlled the coagulation and complement pathway activation even more efficiently than unfractionated heparin and could, therefore, be a potential IBMIR inhibitor in hepatocyte transplantation. Hammel et al. infected hepatocytes with an adenovirus which contained genes encoding the human CR1 receptor, a potent inhibitor of the C3 and C5 convertases, which are crucial in the classical and alternative complement pathways. Hepatocytes expressing the CR1 adenovirus reduced complement activation following transplantation and increased albumin production in immunocompetent Nagase analbuminemic rats [60].

### 6.2. Mesenchymal Stem Cell Transplantation

#### 6.2.1. In vitro Expression of TF and Procoagulant Activity

MSCs from various origins are currently under evaluation in numerous clinical trials. In some clinical trials [3,4,5,6,7], thrombogenic events after MSCs infusions have been described, leading to further investigations of the interaction of MSCs with blood. Stéphenne et al. [61] described that both isolated human adult liver-derived mesenchymal progenitor cells (hALPCs) and bone marrow mesenchymal stem cells (BMMSCs) exhibited significantly measurable procoagulant activity (PCA) by thromboelastometry. This was further confirmed for other cells of mesenchymal phenotype, such as skin fibroblasts and liver myofibroblasts (activated stellate cells). The high PCA of hALPCs was associated with high TF expression and low expression of its natural inhibitor, Tissue Factor Pathway Inhibitor (TFPI), as compared to hepatocytes (PCR, Flow cytometry or immunochemistry). Moll et al. [2,62,63] described that both BMMSCs and placenta-derived decidual stromal cells (DSCs) express a PCA linked to the expression of TF (thromboelastometry, PCR, FACS), inducing, in a whole-blood Chandler tubing model, a decrease in platelets and the formation of thrombin. This PCA is directly linked with the amount of TF expressed by the cells (DCSs > BMMSCs) but is also dose-dependent and increases with ex vivo expansion and cryopreservation. Christy et al. [64] confirmed that the PCA of MSCs is correlated with the percentage of cells expressing surface TF. However, a MSCs population with only 25% of cells expressing TF can already present a significant pro-coagulant stimulus, as evidenced by 70% thromboelastography coagulation time shortening and thrombin generation. Adipose-tissue-derived cells appear to be more pro-coagulant, with higher expression of TF, compared to bone marrow derived cells. Nevertheless, MSCs from the same origin, for example, bone marrow, can vary in the degree of both TF expression and PCA. They also confirmed that handling conditions and growth media likely affected PCA since they observed changes in TF expression over culture time. TF expression and PCA of multiple (bone marrow, adipose tissue, amniotic fluid, umbilical cord, …) human MSCs were also confirmed by many other teams [18,65,66].

In contrast, recently, Netsch et al. [67] published that MSCs from different origins (bone marrow, adipose and cord blood) express anticoagulant properties, affecting primary hemostasis. They showed that MSCS inhibit the agonist-induced activation and aggregation of platelets in platelet-rich plasma (PRP) and whole blood. The underlying mechanism involves CD73-converted adenosine, produced by MSC-platelet co-cultures at inhibitory levels. Adenosine induces an increase in adenosine monophosphate (AMPc) levels and vasodilator-stimulated phosphoprotein (VASP) phosphorylation through different receptors, reducing platelet activation further.

MSCs appear to express both pro- and anti-coagulation properties; however, up to now, only one study has been published on the anti-coagulation properties of MSCs. Further studies are needed to understand the complex interaction between MSCs and coagulation, including the balance between the pro- and anti-coagulation properties expressed by MSCs.

#### 6.2.2. MSCs Infusion in Animal Models

In a mice model, Liao et al. [68] studied the cell-induced coagulation reaction by mouse BMMSC. They demonstrated by tail vein infusion that while increasing the infused cell dose, from 1 to 160 × 10^6^ cells/kg, they induced number of severe symptoms, including dyspnea, cyanosis, tetraplegia, coolness of the extremities and exophthalmos, indicating respiratory and circulatory failure and high mortality. By injecting different concentrations of BMMSC suspensions, they confirmed that it was the total cell number, not the concentration, which determined the acute adverse effects. They performed histological examinations to explore the cause of BMMSC-induced respiratory and circulatory failure. Hematoxylin and Eosin staining showed marked micro-thrombi formation in numerous organs (lungs, heart, liver, kidney, and spleen) and hemostasis analysis revealed significant decreases in platelets, fibrinogen and factor VIIIa in a concentration-dependent manner. This data confirmed that BMMSC activate coagulation after infusion in a cell-dose-dependent manner.

Tatsumi et al. [18] administered high doses, 1.5 × 10^6^ cells per mouse (corresponding to 75 × 10^6^ cells/kg, considering mice weighing 20 g), of mouse adipose-derived MSCs (mADSCs) via tail vein infusion, inducing a mortality rate of nearly 85% within 24 h after injection. When PBS or 10× fewer cells were administered, the mice showed no adverse events and survived. Histological assessment of the mice revealed micro-thrombi in several organs (lung, heart), with infused cells found in the core of the fibrin clots, indicating that administered MSCs triggered thrombus formation around the cells.

Coppin et al. [69], showed that the thrombogenic risk induced by intraportal infusion of human adult liver-derived progenitor cells (HHALPCs) in Wistar rat is cell-dose-dependent. Infusion of high cell doses, such as 50 × 10^6^ cells/kg, induced—after 1 h—a significant decrease in platelets and coagulation factors I, II, V and VIII. By intravital microscopy and immunohistochemistry, alterations in the liver vasculature were observed 24 h after cell infusion, associated with vessel obstruction due to blood clots containing the infused cells. However, liver vascularization normalized spontaneously after 7 days. When they infused 10-times fewer cells, 5 × 10^6^ cells/kg, no changes in blood count levels or liver vascularization were observed.

In a porcine myocardial infarction model, Gleeson et al. [65] found that intra-coronary infusion of pig-derived BMMSCs was associated with a mortality rate of 60%, attributed to exacerbated microvascular obstruction due to platelet-rich thrombi containing infused cells. These results confirmed data from Vulliet et al. [70], showing acute myocardial ischemia and subacute myocardial microinfarction after intracoronary arterial injection of autologous BMMSCs into dogs.

Oeller et al. [71], compared—in a rat model—the infusion (cell doses: 6 × 10^6^ cells/kg) of BMMSC expressing low TF, sorted by FACS, to umbilical-cord-derived MSCs (UCMSCs) with high TF expression. One hour after infusion (by tail vein), pathological examination of the lung, liver and spleen showed massive intravascular thromboembolism for UCMSCs, absent for BMMSC expressing low TF. Confirming the important role of TF in the activation of coagulation by MSC infusion.

#### 6.2.3. MSCs Infusion in Patients

Sokal et al. [5] reported a thrombogenic event during intraportal infusions of adult-derived human liver stem/progenitor cells (ADHLSCs) in a patient presenting with ornithine transcarbamylase deficiency. A partial thrombosis of the left portal vein branch was observed by doppler ultrasound, associated with elevation in D-dimer levels after the second course of infusions. This thrombogenic event led to the arrest of the infusions and the start of an anticoagulant treatment but did not result in any consequences for the patient.

Wu et al. [4] published that two patients with renal transplantation and chronic kidney disease, respectively, experienced thromboembolism after umbilical cord MSC infusion. On the second and third day after infusion, both developed pain and swelling nearby the infusion site. The diagnosis of venous thrombi was made by a decrease in blood flow seen by Doppler ultrasound associated with an elevation in D-Dimer blood levels. Jung et al. [3] reported pulmonary embolisms and infarcts among three patients who received multiple intravenous infusions of autologous adipose tissue MSCs. Others [8] described an increase of thrombin–antithrombin (TAT) and D-dimer levels, both markers of the activation of coagulation, after intravenous infusions of allogeneic adipose MSCs (ASCs).

#### 6.2.4. MSCs and IBMIR

Moll et al. [1,2,72] confirmed and characterized the IBMIR of BMMSCs both in vitro and in vivo. In a Chandler tubing loop model, BMMSCs induced an increased formation of thrombin, coagulation factors VII, XI, and XII and a significant decrease in platelets and PMNs. This was associated with a small increase in C3a and sC5b-9 levels, especially with higher-passage MSCs. However, BMMSCs infusions in 44 patients induced only a weak triggering of IBMIR. A small (~15%) drop in platelets, associated with a five-fold increase in C3a levels without any changes in PMNs, was observed after 24 h.

## 7. Procoagulant Activity Modulation

An important strategy to improve outcomes of MSC therapy and cell survival after transplantation is the modulation of the PCA of MSCs and/or the prevention of the activation of coagulation induced by MSC infusion. Ettelaie et al. [73] showed that low-molecular-weight heparin (LMWH) downregulates TF expression in vitro in tumorous cell lines from different origins (pancreatic, breast, colon, ovarian, and skin cancer). Incubation of these cells with LMWH downregulated TF mRNA expression, causing a progressive reduction of the intracellular TF reservoir, resulting in lower TF activity and release of TF-bearing microparticles levels.

Stéphenne et al. [61] demonstrated, by thromboelastography, that the PCA of hALPCs could not be controlled with antithrombin activator (Heparins, Fondaparinux), direct factor Xa inhibitor (Rivaroxaban) or thrombin inhibitor (Hirudin, Bivalirudin) alone. However, using a combination of anticoagulant drugs, including an antithrombin activator or direct factor Xa inhibitor and a direct thrombin inhibitor controlled the PCA of hALPCs both in vitro and in vivo. They successfully infused two patients using this combination of anticoagulants without observing any thrombogenic or hemorrhagic events. In a larger phase I/II study [38,74], eleven patients presenting metabolic liver diseases (Crigler Najjar and Urea Cycle Disorder) were infused with liver-derived MSCs (HepaStem) using the same combination of anticoagulants, heparin and bivalirudin. All patients presented consumption in platelets and production of d-dimer after cell transplantation, suggesting activation of the coagulation; however, this was spontaneously reversible in time. One patient presented a procoagulant event, such as partial portal vein thrombi, needing additional anticoagulation. No major adverse effects were observed with the use of these anticoagulants during cell infusions.

Recently, Coppin et al. [69], compared two anticoagulant scenarios, i.e., a combination of heparin (10 IU/5 × 10^6^ cells) and bivalirudin previously described [61] and high doses of heparin, 300 I.U./5 × 10^6^ cells. They showed, in a Wistar rat, that the addition of the higher heparin amount was more effective to control the PCA of HHALPCs. Adding high doses of heparin further limited the decrease in platelets and fibrinogen levels and the increase in thrombin generation.

Moll et al. [63] successfully controlled, in vitro in clotting assays, the PCA of BMMSC and DSC by adding low doses of heparin (0.3 U/mL). Although they found a transient increase in D-Dimer levels, no other changes in coagulation or complement parameters were found in eight patients infused with DSCs when cells were washed and reconstituted in buffers containing low-dose heparin and EDTA. They conclude that the rather low amount of heparin and EDTA, which will be diluted in a large amount of blood, transiently prevents the instant formation of aggregates when cells are transferred from the syringe to the blood stream. In a larger study on 44 patients [75], D-Dimers also increased transiently without a decrease in platelets, when DSC’s infusions were preceded and followed with heparin administration (62.5–250 I.U., depending on the weight of the patient), suggesting only a minor activation of the coagulation.

Liao et al. [68], could control the PCA of mouse BMMSC in a clotting assay in vitro by adding heparin (4 U/mL). Nevertheless, adding heparin during cell culture did not reduce TF expression, suggesting that heparin functioned by blocking TF-induced coagulation. In mice, pre-injection or injection of heparin (400 I.U./kg) with mice BMMSCs via the tail vein, prevented all adverse effects after infusion of high cell doses (160–320 × 10^6^ cells/kg), which previously were shown to be lethal in 100% of the infused mice. In a colitis mouse model, adding heparin during infusion of a clinical dose (1 × 10^6^ cells/kg) promoted cell migration, with reduction of the number of cells trapped in the lungs and an increase of cells in the targeted organ, the colon and mesenteric lymph nodes. Heparin treatment also significantly improved the therapeutic effect of the cells in this mice model, by decreasing body weight loss, the disease index and mortality. In control mice, injected only with heparin, this improvement was not found.

In the myocardial infarction model of Gleeson et al. [65] in pigs, the percentage of occluded microvessels in the infarct zone significantly reduced when heparin (200 I.U./kg) was co-administrated with pig BMMSCs. After 24 h, heparin improved BMMSC delivery to the infarct territory, resulting in reduced infarct size and an improved ventricular function 6 weeks post-infarct. Finally, Tatsumi et al. [18] proposed thrombomodulin as a potential anticoagulant agent based on ROTEM and clotting assays.

Nevertheless, modulation of the PCA of MSC by using anticoagulant drugs could influence their homing capacities once infused. Seeger et al. [76] showed, by intravital microscopy, that heparin, but not bivalirudin, impairs the homing of BMMSC in intravenously injected mice in an ear wound and in an infarcted heart model. Heparin blocks SDF-1/CXCR4 signaling by binding to the receptor and the ligand as well, thereby interfering with homing and migration of BMMSCs. Bivalirudin, a thrombin inhibitor, did not interfere with this signaling pathway, suggesting that this anticoagulant drug should be recommended, and not heparin, to prevent activation of coagulation after intracoronary infusion of BMMSCs.

Interestingly, Groeneveld et al. [77] showed that thrombin-dependent intrahepatic fibrin(ogen) deposition, present after partial hepatectomy in mice, contributes to liver generation. It was previously known that platelets play a central role in post-operative liver regeneration, low platelet count is a predictor of delayed liver function recovery and increases post-operative mortality rates after partial hepatectomy in humans. These scientists clearly identified fibrin(ogen) deposition as a central molecule for early intrahepatic platelet accumulation, driving liver regeneration after hepatectomy. Others showed that more components of hemostasis, such as TF and vWF, if selectively removed, delay liver regeneration after liver resection. We could thus hypothesize that activation of coagulation by MSCs, leading to fibrin(ogen) and thus platelet accumulation in the transplanted tissue could be beneficial as well. After MSCs infusion, coagulation should be optimally balanced, to avoid thrombi but allow enough fibrin(ogen) production which could possibly favor tissue regeneration. Further studies are needed to investigate the role of activation of coagulation induced by MSCs infusion on cell engraftment and subsequent tissue regeneration.

## 8. Conclusions

MSCs are used in numerous clinical trials nowadays, but their blood compatibility is still an important concern for health professionals. Different studies showed—both in vitro and in vivo—that MSCs express a procoagulant activity, linked to the expression of TF, leading to thrombi formation after blood exposure. However, anticoagulant drugs, such as heparin and/or bivalirudin could prevent (in vitro) and limit (in vivo) the activation of coagulation by these cells. Further studies, nevertheless, are needed to completely control the activation of coagulation after MSC infusion in patients, which would most likely enhance their therapeutic effect.

## Figures and Tables

**Figure 1 cells-08-01160-f001:**
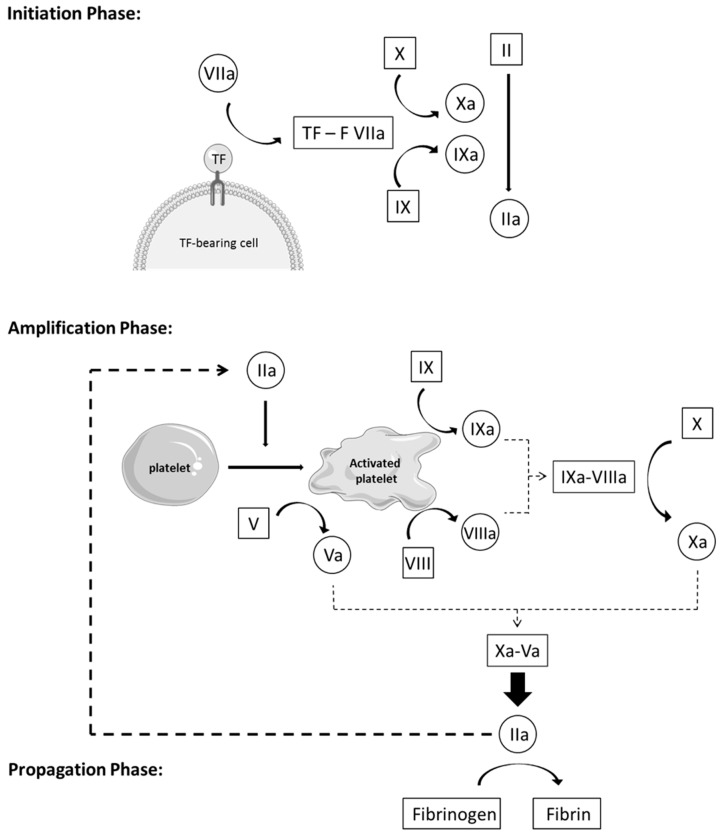
Activation of the coagulation can be described in three phases: During the initiation phase, small amounts of thrombin (IIa) are generated, inducing an important amplification loop through activation of platelets and cofactors V and VIII during the amplification phase. A large amount of thrombin (IIa) is thus generated, leading to an efficient burst of all enzymatic reactions involved in the entire coagulation process. In turn, during the propagation phase, these vast amounts of thrombin (IIa) lead to large quantities of fibrin strains, the final product of coagulation, consolidating the blood clot.

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
