# Peer review of "Thrombogenic Risk Induced by Intravascular Mesenchymal Stem Cell Therapy: Current Status and Future Perspectives"

_cells, 2019, doi:10.3390/cells8101160_

Round 1
Reviewer 1 Report
Cell-based therapies are gaining importance in daily medical practice. Donor (allogenic) cells are infused and delivered commonly through circulatory system. Such non-physiological delivery route, for cells as epithelial and stromal, have been shown to generate thrombosis and massive loss of donor cells. IBMIR has been recently proposed as major cause for such cell loss, with activation of complement and coagulation pathways primarily involved.
Coppin and coauthors provide an excellent overview both for the coagulation cascade and the current cellular therapies (and related IBMIR effect). Another exhaustive review manuscript has been previously published (Cell Transplantation 2016;25:1227-36) by a different group. The current manuscript similarly described IBMIR in vitro and invivo analysis. Larger and accurate analysis is provided by Coppin and coauthors on stromal cells infusions, in particular on human cell products the authors have been actively manufactured and infused during the past 5 years.
A critical analysis and therapeutic overview has been provided, with important description of pancreatic islets and liver cell infusions. Nevertheless, limited description of the hepatic therapeutic approaches has been unfortunately provided. The authors are experts in such area, thus a more detailed and punctuate analysis of current limitations and proposed solutions would largely benefit the manuscript.
Comments on preclinical models, originally tested with transplanted liver cells and complement activation (Hammel et al. Transpl proc 1999;31(98):939) are encouraged.
Excellent studies have been published by a Swedish group in Uppsala, however only one paper (the first one published in 2011; ref 35) has been cited. Two high-quality manuscripts have been recently published describing IBMIR on liver cells (Cell Transplant. 2017;26(1):71-81; Transplantation. 2019;103(8):1630-1638) and they should be included and discussed.
Limited discussion on therapeutic strategies proposed to inhibit IBMIR, such as the addition of NAC (N-acetylcysteine), or macromolecular heparin complexes, or low-molecular-weight dextran sulfate, or melagatran, a direct thrombin inhibitor described to reduce thrombinformation through the production of chondroitin sulphate (Cabric et al. Tissue Eng part C 2008;14(2):141; Ozmen et al. Diabetes 2002;51(6):1779 ). Please motivate.
Supportive figures, such as a diagram to illustrate IBMIR in vitro analysis, would largely benefit reader comprehension.
Author Response
Response to Reviewer 1 Comments
Cell-based therapies are gaining importance in daily medical practice. Donor (allogenic) cells are infused and delivered commonly through circulatory system. Such non-physiological delivery route, for cells as epithelial and stromal, have been shown to generate thrombosis and massive loss of donor cells. IBMIR has been recently proposed as major cause for such cell loss, with activation of complement and coagulation pathways primarily involved.
Coppin and coauthors provide an excellent overview both for the coagulation cascade and the current cellular therapies (and related IBMIR effect). Another exhaustive review manuscript has been previously published (Cell Transplantation 2016;25:1227-36) by a different group. The current manuscript similarly described IBMIR in vitro and invivo analysis. Larger and accurate analysis is provided by Coppin and coauthors on stromal cells infusions, in particular on human cell products the authors have been actively manufactured and infused during the past 5 years.
Point 1: A critical analysis and therapeutic overview has been provided, with important description of pancreatic islets and liver cell infusions. Nevertheless, limited description of the hepatic therapeutic approaches has been unfortunately provided. The authors are experts in such area, thus a more detailed and punctuate analysis of current limitations and proposed solutions would largely benefit the manuscript.
Response 1: We thank the author for this interesting comment. An extended chapter on MSCs transplantation has been added in section 4. We added a paragraph specifically about the application of MSC transplantation in liver diseases, sharing the possible applications, problems and prospects like the reviewer suggested.
Due to their immunomodulatory and regenerative properties, MSCs can potentially be used as a novel therapy for many diseases. MSCs can for example be used for the treatment of liver diseases which is a major health concern with important morbidity and mortality. In general, all types of chronic hepatitis will without an efficient treatment progress to end-stage liver disease, for which the only curative treatment is liver transplantation [31]. However due to organ shortage, other therapies such as liver MSCs transplantations are currently investigated. Promising results were first obtained for patients presenting inborn metabolic liver diseases [3,32], such as Crigler-Najjar syndrome or urea cycle defects, but long term cell engraftment and graft function however are still lacking. The immunomodulatory and antifibrotic properties of different types of MSCs were then investigated to treat liver cirrhosis [31] and acute liver failure [33]. Recently, encouraging results were obtained using liver-derived MSCs in cirrhotic patients with acute-on-chronic liver failure or acute decompensation, inducing a significant MELD and bilirubin improvement [34].
Point 2: Comments on preclinical models, originally tested with transplanted liver cells and complement activation (Hammel et al. Transpl proc 1999;31(98):939) are encouraged.
Response 2: We would like to thank the reviewer for giving us the opportunity to read this very interesting and encouraging article. The following has been added in section 4.1.:
Hammel et al. infected hepatocytes with an adenovirus which contained genes encoding the human CR1 receptor, a potent inhibitor of the C3 and C5 convertases, which are crucial in the classical and alternative complement pathway. Hepatocytes expressing the CR1 adenovirus reduced complement activation following transplantation and increased the albumin production in immunocompetent Nagase analbuminemic rats [43]
Point 3: Excellent studies have been published by a Swedish group in Uppsala, however only one paper (the first one published in 2011; ref 35) has been cited. Two high-quality manuscripts have been recently published describing IBMIR on liver cells (Cell Transplant. 2017;26(1):71-81; Transplantation. 2019;103(8):1630-1638) and they should be included and discussed.
Response 3: We thank the reviewer for informing us about these two papers. We already read extensively the very interesting thesis of E. Gustafson and this Swedish group in Uppsala on IBMIR induced by hepatocytes. The two proposed articles were added in the review and discussed in section 4.1.:
Recently, the same group showed that the expression of vWF on the surface of isolated hepatocytes could induce, in contact with blood, adhesion of platelets forming an ideal surface for coagulation. Once thrombin is generated, the thromboinflammatory reactions will then be strongly amplified and cause a rapid destruction of the exposed cells [43].
Different strategies have been investigated to prevent IBMIR after hepatocytes transplantation. Gustafson et al.[44], compared the capacity of low molecular weight dextran sulfate (LMW-DS) to conventional anticoagulant therapy, unfractioned heparin, to control IBMIR in vitro in a whole blood Chandler loop model. They showed that LMW-DS controlled the coagulation and complement pathway activation even more efficiently then unfractioned heparin and could therefore be a potential IBMIR inhibitor in hepatocyte transplantation.
Point 4: Limited discussion on therapeutic strategies proposed to inhibit IBMIR, such as the addition of NAC (N-acetylcysteine), or macromolecular heparin complexes, or low-molecular-weight dextran sulfate, or melagatran, a direct thrombin inhibitor described to reduce thrombinformation through the production of chondroitin sulphate (Cabric et al. Tissue Eng part C 2008;14(2):141; Ozmen et al. Diabetes 2002;51(6):1779 ). Please motivate.
Response 4: We thank the reviewer for these kind remarks. We wanted in this review to mainly focus on MSCs transplantation. Hepatocyte and islets transplantation was more used as an introduction to the subject. However, these interesting articles were added to inform the readers on potential strategies to prevent IBMIR in hepatocytes/islets transplantation in section 4.1.:
Cabric et al.[39] attempted to prevent IBMIR by applying surface-immobilized functional heparin to islets, trying to mimic the protective anticoagulant activity normally present at the endothelial lining of the vascular wall. They showed that the entire cell surface of the cells could be covered with heparin and did not interfere with the function of the islets. Identical insulin secretion and engraftment rats were observed in a syngeneic islet transplantation model in diabetic mice. Others successfully reduced IBMIR by adding a thrombin inhibitor, Melagatran [40], activated protein C synergistically with a platelet glycoprotein IIb‐IIIa inhibitor, Tirofiban [41], N-acetyl-L-cysteine [42] or even α-1 antitrypsin [43].
Point 5: Supportive figures, such as a diagram to illustrate IBMIR in vitro analysis, would largely benefit reader comprehension.
Dear reviewer, we already added numerous studies about IBMIR in the paper. As it is not the main focus of the review, we ask to not add a supplementary figure. We think that IBMIR has been extensively reviewed in the article by Moll. Et al 2019.
Reviewer 2 Report
In this paper, the authors review studies assessing thrombogenic risk associated with intravascular mesenchymal stem cell therapy. This subject has already been reviewed in a fairly recent article [2] but an update is justified by the abundance of recent studied addressing the subject. Although I think the work deserves to be published in Cells, there are a number of points that need clarification prior to publication. I will start with the overall comments.
My main concern is about the structure of the manuscript. I found that the content of the manuscript does not well reflect the topic of the review. Many details provided in section 2 are not used at all in the following sections, while too little is said about MSC and the protocols used in MSC therapy (origin of MSC, what type of cells for which therapeutic goal, injection sites, culture media, cell quantities). It would be nice to provide a few general facts about MSC therapy before discussing its adverse effects. Also, the authors seem to assume that the reader is familiar with the complement pathway, knows what C3a is and how to interpret an increase in C3a. Given the wide scope of the journal, I think all these points should be reminded or explained in a more explicit manner. The introduction is extremely brief and in my opinion fails to set the general context of the review. The last sentence in particular is way too vague and conveys the incorrect impression that the phenomena are studied at the scale of a single cell. I think the introduction should be rewritten in a more precise manner, detailing what is at stake in this review and announce its structure. The same holds for the discussion: what heave we learned since ref [2] ? The discussion about anticoagulant treatments is supported by data from patients, but also from different animal models (pig, rat, mouse). To what extent is the coagulation cascade presented in the first part of the paper conserved across species ?In addition, I have a few more minor or specific remarks or questions
The coagulation cascade is defined in line 65 as « the generation of fibrin by activating inactive soluble factors trough a complex proteolytic cascade ». This seems in fair agreement with what is sketched in figure 1 and described as « activation of coagulation ». Could the authors better explain the difference between « activation of coagulation » and « the coagulation cascade », or else provide more consistent names ? Refs [45] and [54] lead to contrasted conclusions about the effect of bivalirudin on the procoagulant activity of MSC. The two studies differ by many parameters. Among them (animal model, cell type, injection site), which is/are the most likely to trigger these different responses ? In section 4.2.1 the authors discuss the pro- and anti-coagulant activities of MSC. It would be nice to add one or two concluding sentences to discuss the balance between these two opposite effects.Finally, though the manuscript is generally understandable, some sentences are poorly written. There are sentences without verb and/or using unappropriate tenses (line 90/91 for instance). I would recommend to work with a native English reader to improve the quality of the language.
Author Response
Response to Reviewer 2 Comments
In this paper, the authors review studies assessing thrombogenic risk associated with intravascular mesenchymal stem cell therapy. This subject has already been reviewed in a fairly recent article [2] but an update is justified by the abundance of recent studied addressing the subject. Although I think the work deserves to be published in Cells, there are a number of points that need clarification prior to publication. I will start with the overall comments.
Point 1:
My main concern is about the structure of the manuscript. I found that the content of the manuscript does not well reflect the topic of the review. Many details provided in section 2 are not used at all in the following sections, while too little is said about MSC and the protocols used in MSC therapy (origin of MSC, what type of cells for which therapeutic goal, injection sites, culture media, cell quantities). It would be nice to provide a few general facts about MSC therapy before discussing its adverse effects.
Response 1: We agree completely with the reviewer and added a section on cell-based therapy and especially on MSCs (section 5): see line 199 to 249.
Point 2: Also, the authors seem to assume that the reader is familiar with the complement pathway, knows what C3a is and how to interpret an increase in C3a. Given the wide scope of the journal, I think all these points should be reminded or explained in a more explicit manner.
Response 2: We thank the reviewer for this interesting remark. A section on the cross-talk between coagulation and inflammation has been added, where the complement pathway is explained briefly. We found it more interesting to explain the link between coagulation and inflammation then explain the complement pathway fully: see line 146 to 174.
Point 3: The introduction is extremely brief and in my opinion fails to set the general context of the review. The last sentence in particular is way too vague and conveys the incorrect impression that the phenomena are studied at the scale of a single cell. I think the introduction should be rewritten in a more precise manner, detailing what is at stake in this review and announce its structure.
Response 3: I would like to thank the reviewer for this on point remark. However, we didn’t want to repeat ourselves in this review and preferred to have a rather shorter introduction and give more details subsequently in the different sections. Nevertheless, the structure and the aim of this review has been clarified. The introduction has been rewritten as followed:
Solid organ transplantation is still the treatment of choice for patients presenting end-stage organ disease. However due to organ shortage, the growing demand for organs and the substantial risks of morbidity and mortality linked to immunosuppression and surgery, other treatments such as cell-based therapy are being developed. The aim is that the infused cells migrate to the injured tissue to regrow, repair or replace the diseased cells. Even though, mature cells were first investigated, nowadays there is a growing interest for the use of mesenchymal stromal cells (MSCs), because they can easily be isolated and expanded in vitro, differentiate into various cell lineages and present immunomodulatory properties. However, some concerns have been raised regarding their safety and their compatibility with blood [1,2]. Activation of coagulation has indeed been reported in several transplanted patients [3-8], and could be linked to tissue factor expression. Some even describe a simultaneous activation of the complement pathway, called the Instant Blood Mediated Inflammatory Reaction (IBMIR), that could explain the low engraftment rates of the transplanted cells.
To further optimize MSCs transplantation, the interaction between the transplanted cells and the injection site, being often the blood circulation, has to be well understood. Understanding the mechanism, gives the opportunity to develop novel targets/strategies to prevent the activation of both the coagulation and complement pathway. Therefore, in this review, we first summarize briefly the concept of hemostasis, the effect of tissue factor and the cross-talk between coagulation and inflammation. Next, we discuss current knowledge about the interaction between transplanted cells and blood, specifically focusing on the thrombogenic risk induced by MSCs. We discuss the link between coagulation and inflammation through the IBMIR. And finally, we review the different methods, that groups investigated to modulate this thrombogenic risk and improve MSCs therapy.
Point 4: The same holds for the discussion: what heave we learned since ref [2] ? Moll?
Response 4: Even though our review has some similarities with the one written by Moll, the focus of both reviews is however different. Moll et al. mainly reported on the hemocompatibility of MSCs triggering adverse innate immune responses, focussing on the use of bone marrow MSCs comparing them to MSC from other sources.
In contrast, our review mainly focussed on the procoagulant activity of MSCs from different sources. We reviewed the different mechanisms describes in literature, the difference between the in vitro and in vivo results, and the different methods to control activation of coagulation induced by MSCs infusion. We also shared our knowledge on treating liver diseases with MSCs.
Point 5:
The discussion about anticoagulant treatments is supported by data from patients, but also from different animal models (pig, rat, mouse). To what extent is the coagulation cascade presented in the first part of the paper conserved across species ?
Response 5: We thank the reviewer for this good remark. The coagulation system is indeed different between species, even though some similarities can be found, and the same laboratory tests can be used (Siller-Matula Thromb. Haemost. 2008, Garcia-Manzano Proceedings of the Western Pharmacology Society 2001, Kaibara J. of Japanese Society of Biorheology 2006). However, like in the development of most novel therapies, in vivo animal studies have to be performed first to investigate the safety and efficacy of the treatment. Most studies that we cited however described the procoagulant activity of their MSCs both in animal studies and in in vitro studies using human blood (Chandler tubing loop models).
Point 6: In addition, I have a few more minor or specific remarks or questions
The coagulation cascade is defined in line 65 as « the generation of fibrin by activating inactive soluble factors trough a complex proteolytic cascade ». This seems in fair agreement with what is sketched in figure 1 and described as « activation of coagulation ». Could the authors better explain the difference between « activation of coagulation » and « the coagulation cascade », or else provide more consistent names ?
Response 6 : We would like to thank the reviewer to have noticed this inconsistency. Both denominations are used in the literature and have the same meaning. To provide more consistency, we used the activation of coagulation instead of the coagulation cascade. This has been adapted through the whole manuscript.
Refs [45] and [54] lead to contrasted conclusions about the effect of bivalirudin on the procoagulant activity of MSC. The two studies differ by many parameters. Among them (animal model, cell type, injection site), which is/are the most likely to trigger these different responses ?
We thank the reviewer for this remark, but we think these references do not lead to different conclusions regarding bivalirudin.
Seeger et al. studied the effect of anticoagulants, heparin and bivalirudin on the homing capacities of bone marrow derived mononuclear cells. They showed that heparin, but not bivalirudin interfered with the homing capacities of these cells through the CXCR4/SDF1 axis.
Our article however studied if anticoagulants, such as heparin and bivalirudin, could prevent the activation of coagulation induced by liver derived MSCs. The effect of these anticoagulants on homing capacities has however never been studied in this article. Most investigations focused on the effect on coagulation 1h after cell infusion, so nothing can be said about homing capacities. Besides, one of the remarks in the article is that this has to indeed still be investigated.
In section 4.2.1 the authors discuss the pro- and anti-coagulant activities of MSC. It would be nice to add one or two concluding sentences to discuss the balance between these two opposite effects.
We would like to thank the reviewer for this suggestion. Up to now only 1 study has been published on the anticoagulation properties of MSCs, however we are convinced that this is a real interesting topic that needs to be explored in future studies. Following concluding sentences have been added:
MSCs seem to express both pro- and anti-coagulation properties, however, up to now, only one study has been published on the anti-coagulation properties of MSCs. Further studies are needed to understand the complex interaction between MSCs and coagulation, including the balance between pro- and anti-coagulation properties expressed by MSCs.
Finally, though the manuscript is generally understandable, some sentences are poorly written. There are sentences without verb and/or using unappropriate tenses (line 90/91 for instance).
The sentence, previously at line 90/91 has been rewritten.
I would recommend to work with a native English reader to improve the quality of the language.
The article has been edited by the editing services from MDPI.
Round 2
Reviewer 2 Report
In this modified version, the authors fixed the main faults of the manuscript. I think it now meets the standards of the journal.